# Exploring Perturbation Effects on Transcriptional Dynamics with ContrastiveBiVI

**Ashenafee A. Mandefro**[*]
Department of Computer Science
University of Toronto
Toronto, Canada
`ashenafee@cs.toronto.edu`

**Ethan Weinberger, Addie Woicik & Tal Ashuach**
insitro
South San Francisco, CA 94080, USA
`{ethan.weinberger,addiewc,tal.ashuach}@insitro.com`

## Abstract

CRISPR-Cas9-based genetic screens combined with single-cell transcriptomic profiling have emerged as a powerful tool for investigating the relationships between genotype and cellular phenotypes. However, most computational analyses of these screens solely focus on comparisons of observed measurements of mature mRNA abundance between groups of cells, and ignore the underlying biophysical phenomena (e.g., mRNA splicing dynamics) that lead to the observed data. Towards explicitly modelling such phenomena in single-cell RNA sequencing data, a recent line of work has proposed simultaneously considering nascent and mature mRNA count levels and relating the two via biophysically plausible models of transcription formalized by chemical master equations. Yet, applying these models directly to perturbation screening datasets is not straightforward, as perturbation-induced variations of interest are often confounded by uninteresting "background" variation shared with control cells. To remedy this issue, here we propose ContrastiveBiVI, a generative model that combines ideas from the biophysical modelling literature with so-called contrastive latent variable models, which explicitly disentangle perturbation-induced variations from non-perturbation-related variations into separate sets of latent factors. Applied to a publicly available CRISPR activation dataset, we find that our method successfully recovers perturbation-induced variations and facilitates the exploration of perturbation-induced changes in transcription and splicing kinetics. An open-source implementation of our model is available at `https://github.com/insitro/contrastive_bivi`.

## 1 Introduction

Advances in high-throughput single-cell RNA sequencing (scRNA-seq) protocols have enabled transcriptome-wide measurements of tens of thousands to millions of cells in a single experiment. Moreover, pooled genetic screening protocols combining CRISPR-Cas9-mediated genome editing with high-content single-cell readouts, such as Perturb-seq (Dixit et al., 2016), hold major promise for identifying the causal bases of cellular phenotypes. In particular, a central goal of collecting such datasets is to uncover how the processes of transcriptional regulation lead to observed cellular phenotypes. However, the majority of computational analyses (Replogle et al., 2022; Norman et al., 2019; Papalexi et al., 2021) of single-cell genetic perturbation datasets have primarily focused on differences in mature mRNA abundance between conditions. While such analyses have led to new biological insights, they elide the underlying biophysical mechanisms (e.g., mRNA splicing dynamics) that give rise to observed perturbation effects.

---

[*]Work performed during an internship at insitro.

To enable a deeper understanding of the regulatory phenomena underlying observed scRNA-seq data, a recent line of work (Gorin et al., 2023; Carilli et al., 2024; Gorin et al., 2025) has proposed considering both nascent and mature mRNA counts and relating the two via biophysically plausible generative models of transcription formalized by chemical master equations (CMEs). Such approaches can enhance analyses of scRNA-seq data by equipping resulting model parameters with direct biophysical interpretations. Moreover, these approaches can elucidate differences in transcriptional modulation between groups of cells, even when no differences are present in mean spliced RNA abundance levels alone (Carilli et al., 2024).

Despite the promise of these biophysically-grounded modelling approaches, applying them specifically to analyze perturbation screening datasets comes with an additional set of challenges. In particular, perturbation-induced variations of interest in single-cell perturbation screens may be subtle compared to those due to other biological processes, such as cell-cycle-related variations or those due to cellular stress responses (Papalexi et al., 2021; Weinberger et al., 2023; 2024). Thus, standard latent variable modelling frameworks (e.g., based on variational autoencoders (Kingma & Welling, 2013; Lopez et al., 2018)) that use a single set of latent factors to capture all of the variation in a dataset may fail to capture perturbation effects in scRNA-seq data, as these methods prioritize capturing factors with the highest variance across an entire dataset; as demonstrated in our results, this phenomenon continues to hold true even when such models are formulated with a biophysically grounded generative process.

In parallel, a separate line of work has formulated so-called contrastive latent variable models (cLVMs) based on the principle of contrastive analysis (Zou et al., 2013). Such models explicitly disentangle perturbation-induced variations into a set of *salient* latent variables while factors of variation present in both control and perturbed samples are segregated into a second set of *background* variables. While previously proposed cLVMs (Weinberger et al., 2023; 2024; Jones et al., 2021; Lopez et al., 2024) have demonstrated promise for analyzing pooled CRISPR screening datasets, they have focused exclusively on modelling mature mRNA abundance, while ignoring the underlying biophysical processes that regulate the mRNA lifecycle.

To enable deeper explorations of rich scRNA-seq perturbation datasets, here we propose ContrastiveBiVI, a biophysically-grounded cLVM that facilitates both the isolation of perturbation-induced effects in scRNA-seq genetic perturbation screening data and the biophysical interpretation of perturbation effects on mRNA splicing dynamics. The remainder of this work proceeds as follows. In Section 2 we review CME-based approaches for analyzing scRNA-seq data along with cLVMs for analyzing perturbation datasets. We then proceed (Section 3) to describe our proposed model's generative process and (Section 4) its corresponding inference procedure. Finally, in Section 5 we apply our method to re-analyze a publicly available CRISPR activation (CRISPRa) screen on K562 cells originally presented in Norman et al. (2019). We find that our model successfully isolates perturbation-induced effects in this dataset, and can identify changes in splicing kinetics induced by perturbations even when no changes are observed in mature mRNA abundance alone.

## 2 BACKGROUND

### 2.1 CONTRASTIVE LATENT VARIABLE MODELS

Contrastive analysis (CA) seeks to separate sources of variation that are specific to a condition of interest from those that are common across conditions. To achieve this objective, several recent works (Abid & Zou, 2019; Severson et al., 2018; Weinberger et al., 2023; Jones et al., 2021) have introduced contrastive latent variable models (cLVMs) using the following framework. Consider a perturbed observation $x_i$ (e.g., a cell infected with a non-control CRISPR guide RNA). We posit that $x_i$ is generated from some random process with parameters $\theta$, conditioned on two disjoint sets of latent variables $z_i$ and $t_i$:

$$x_i \sim p_\theta(x_i \mid z_i, t_i),$$

In our construction, we assume that $t_i$ constitutes the set of *salient* latent variables responsible for the unique variations present only in perturbed samples. Conversely, $z_i$ encodes *background* variations, corresponding to phenomena present in both the control and experimental groups, and which are not of interest for our analysis. Importantly, if one were to simply fit such a model using standard latent variable inference techniques, there is no reason to expect that the inference procedure would

correctly segment the shared and perturbation-specific signals cleanly to $z_i$ and $t_i$, respectively, rather than distributing them arbitrarily across both sets of latents.

To encourage this desired separation, cLVMs inject information from our corresponding control samples as an inductive bias. Concretely, for a control observation $x_j^{\varnothing}$ we assume that it is generated by the same generative process $p_\theta$, but with the salient component $(t_j)$ fixed to 0 to represent the absence of perturbation-specific effects

$$x_j^{\varnothing} \sim p_\theta(x_j^{\varnothing} \mid z_i, 0).$$

That is, during inference the model is constrained such that control samples must be explained solely through the background latents $z$, while perturbed samples may additionally use the salient latents $t$. This asymmetry implicitly encourages variation that is common to both groups to be encoded in $z$, leaving $t$ to capture what is present only in the perturbed population. Using this idea, previous works have successfully demonstrated the ability to successfully isolate perturbation-induced variations in scRNA-seq data from those shared with controls using both linear (Jones et al., 2021) and nonlinear, deep-neural-network-based models (Weinberger et al., 2023). Nevertheless, as discussed previously, prior cLVMs for scRNA-seq are not suitable for understanding the mechanistic effects of genetic perturbations, as they exclusively consider mature mRNA abundance.

## 2.2 Mechanistic Generative Models of Transcriptional Dynamics

Mechanistic models of transcription begin by positing a chemical reaction describing the production and degradation of given molecular species of interest over time. For example, substantial experimental evidence (Suter et al., 2011; Raj et al., 2006) suggests that mRNA molecules are transcribed from DNA in instantaneous "bursts" of multiple copy numbers before gradually decaying over time. For a given gene, we could then approximate the behaviour of its mRNA molecules $\mathcal{X}$ by the reaction

$$\emptyset \xrightarrow{k} B \times \mathcal{X} \xrightarrow{\gamma} \emptyset, \tag{1}$$

where $k$ denotes a constant mRNA production rate, $\gamma$ is a corresponding constant degradation rate, and $B$ is a geometrically distributed random variable with mean $b$ representing the number of molecules generated during a transcriptional event.

From a given reaction, a corresponding master equation - i.e., a differential equation describing how the distribution of the reaction's molecular species vary over time - can then be derived. In the case of Equation (1), the implied master equation can be solved analytically to find that the marginal distribution of mRNA counts at steady state $P(x)$ follows a negative binomial distribution with shape $k/\gamma$ and scale $b$ (Phillips et al., 2012). Notably, this result provides a mechanistic motivation for the negative binomial likelihood commonly employed in generative models of scRNA-seq data, such as in the popular variational-autoencoder-based scVI model (Lopez et al., 2018).

Typical single cell analyses only consider the mature transcriptome, i.e., reads that map to exonic regions of DNA. However, with standard scRNA-seq assays it is often possible to realign raw sequencing outputs to an expanded intron-annotated reference, thereby resulting in two count matrices: a "standard" mature mRNA matrix as well as a nascent mRNA matrix containing counts associated with intronic regions. For a given gene, we can exploit this additional information by positing a richer two-species reaction that accounts for mRNA splicing in addition to transcription and degradation. For example, the single species bursty model in Equation (1) can be extended to

$$\emptyset \xrightarrow{k} B \times \mathcal{N} \xrightarrow{\beta} \mathcal{M} \xrightarrow{\gamma} \emptyset, \tag{2}$$

where $\mathcal{N}$ and $\mathcal{M}$ denote nascent and mature mRNA, respectively, and $\beta$ denotes a (constant) splicing rate. This two species reaction corresponds to a CME that induces a bivariate probability law $P_{\text{Bursty}}(n, m)$ describing the distribution of both types of mRNA at steady state. Such CME-derived bivariate probability laws can be employed as conditional likelihood functions within deep generative models to jointly model the two types of counts while accounting for known causal relationships between them, as done in the biVI model of Carilli et al. (2024), which extends scVI to model both types of counts in a principled manner. However, doing so requires some additional effort.

In particular, while the nascent marginal distribution $P_{\text{Bursty}}(n)$ implied by Equation (2) can be shown to follow a negative binomial distribution, the joint distribution $P_{\text{Bursty}}(n, m)$ and conditional

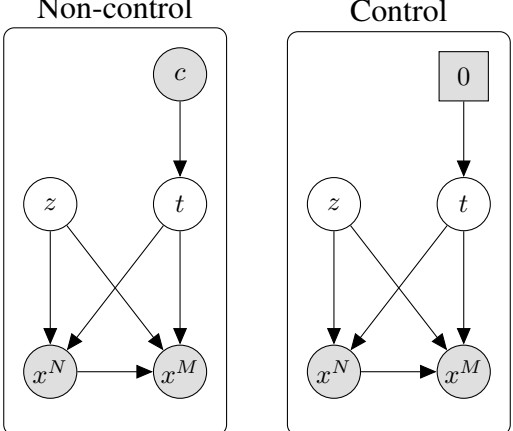

Figure 1: Graphical representation of the ContrastiveBiVI generative process for cells with non-control guides (left) and control guides (right).

distribution $P_{\text{Bursty}}(m \mid n)$ are analytically intractable. To work around this issue, Carilli et al. (2024) proposed parameterizing the conditional likelihood using a pre-trained neural function approximator trained on numerical steady-state solutions of the corresponding chemical master equation (Gorin et al., 2024). In particular, $P_{\text{Bursty}}(m \mid n)$ is approximated using a mixture of negative binomial basis distributions, whose mixture weights are produced by a neural network as a function of the biophysical parameters $(b, \beta/k, \gamma/k)$. This network is trained offline by minimizing the Kullback–Leibler divergence to numerical CME solutions and is held fixed during subsequent generative model training. By doing so, the resulting inferred conditional likelihood parameters can be interpreted as biophysical values parameterizing causal relationships according to a given model of transcription.

## 3 GENERATIVE PROCESS

For a cell $i$ expressing a non-control guide RNA $c_i$, let

$$z_i \sim \mathcal{N}(0, I_p)$$

designate a set of lower-dimensional background latent variables corresponding to factors of variation common to both perturbed cells as well as controls. Next, let

$$t_i \mid c_i \sim \mathcal{N}(\mu_c, I_q)$$

denote a second set of salient latent variables corresponding to variations unique to perturbed cells. In particular, we assume that a perturbed cell's salient latent variables are drawn from a Gaussian centred at a perturbation-specific mean $\mu_c$. For all results in this work we set our perturbation labels $c_i$ as the gene targeted by the expressed guide in cell $i$, although in principle labels at other levels of granularity could be used (e.g., $c_i$ could denote the specific gRNA species expressed in a cell in the case that multiple guides targeting the same gene are used in an experiment).

Letting $f^\eta$ denote a neural network with parameters $\eta$, we then compute

$$\rho_i^N, \rho_i^M = f^\eta(z_i, t_i),$$

where the lengths of $\rho_i^N$ and $\rho_i^M$ are both equal to the number of measured gene features. A softmax activation is then applied to this pair of vectors such that the sum of the entries of $\rho_i^N$ and $\rho_i^M$ equals 1. Analogous to scVI (Lopez et al., 2018), these two vectors represent the expected normalized expression frequency of each gene $g$ for nascent ($\rho_i^N$) and mature ($\rho_i^M$) counts.

For a gene $g$ we then assume that nascent and mature mRNA counts are generated via the two-species bursty model of transcription of Equation (2). Let $\ell_i$ denote the observed library size for cell

$i$. We then assume that the observed number of nascent mRNA counts $x_{ig}^N$ is drawn from a Negative Binomial marginal distribution:

$$x_{ig}^N \mid z_i, t_i \sim \text{NB}\left(\mu_{ig}^N, \alpha_g\right),$$

where $\mu_{ig}^N = \ell_i \rho_{ig}^N$. Conditioned on the nascent mRNA count value for a given gene along with our latent variables, the mature mRNA counts $x_{ig}^M$ are then drawn as

$$x_{ig}^M \mid x_{ig}^N, z_i, t_i \sim P_{\text{Bursty}}\left(m \mid n = x_{ig}^N; \mu_{ig}^N, \mu_{ig}^M, \alpha_g\right)$$

where $P_{\text{Bursty}}(m \mid n)$ denotes the conditional distribution derived from the steady state joint distribution of the two-species bursty transcriptional model, $\mu_{ig}^M$ is computed as $\ell_i \rho_{ig}^M$, and $\alpha \in \mathbb{R}^G$ is a vector of per-gene free parameters jointly optimized across all cells. As discussed in Section 2.2, while this conditional distribution does not admit a closed form, it can be approximated by a mixture of negative binomial distributions whose weights are output by a neural-network-based solver compatible with automatic differentiation. As in Carilli et al. (2024), the per-gene biophysical parameters of our posited reaction can then be recovered by assuming the relation $\alpha = k/\beta$. In particular, the burst size parameter $b$ and relative degradation rate $\gamma/k$ can then be recovered via

$$b_{ig} = \frac{\mu_{ig}^N}{\alpha_g}, \quad \text{and} \quad \frac{\gamma_{ig}}{k_{ig}} = \frac{\mu_{ig}^N}{\alpha_g \mu_{ig}^M}. \tag{3}$$

For a cell $j$ infected with a control gRNA, we assume the same generative process but with $t_j$ drawn from a delta distribution centred at $0$. Thus, semantically the region around $0$ in the salient latent space corresponds to the absence of perturbation-induced variations.

Our generative process closely follows that of biVI (Carilli et al., 2024), with the addition of the salient latent factors $t$. While biVI is suitable for many biophysically based analyses of scRNA-seq, we found (Section 5) that it was not suited for exploring perturbation-induced variations out of the box. We depict our generative process for cells with control and non-control gRNAs in graphical model form in **Fig. 1**.

## 4 INFERENCE

Exact posterior inference for our model is intractable, so we instead resort to variational inference (Blei et al., 2017) via auto-encoding variational Bayes (Kingma & Welling, 2013). For cells with non-control guides, we assume that our variational distribution with parameters $\phi$ factorizes as follows

$$q_\phi(z_i, t_i \mid x_i^N, x_i^M) = q_{\phi_z}(z_i \mid x_i^N, x_i^M) q_{\phi_t}(t_i \mid x_i^N, x_i^M)$$

where $\phi_z$, and $\phi_t$ denote parameters of inference networks for $z$ and $t$, respectively. Here $q(z_i \mid x_i^N, x_i^M)$ and $q(t_i \mid x_i^N, x_i^M)$ take the form of Gaussian distributions. Our corresponding variational bound is then (derivation in Appendix A):

$$\begin{aligned}
\mathcal{L}(x_i^N, x_i^M) = &\mathbb{E}_{q_{\phi_z}(z_i \mid x_i^N, x_i^M) q_{\phi_t}(t_i \mid x_i^N, x_i^M)} \left[\log p_\theta(x_i^N, x_i^M \mid z_i, t_i)\right] \\
&- D_{KL}(q_{\phi_z}(z_i \mid x_i^N, x_i^M) \| p(z_i)) - D_{KL}(q_{\phi_t}(t_i \mid x_i^N, x_i^M) \| p(t_i \mid c_i))
\end{aligned} \tag{4}$$

For cells with non-targeting control (NTC) guides, we assume an alternative variational distribution incorporating our prior knowledge that factorizes as

$$q_{\phi_{NTC}}(z_j, t_j \mid x_j^N, x_j^M) = q_{\phi_z}(z_j \mid x_j^N, x_j^M)\delta\{t_j = 0\}$$

That is, for control cells we assume that $t_j$ is fixed at $0$ to reflect the fact that cells with NTC guides are known to be unperturbed and that the salient variables $t_j$ should not capture variations in the observed data for control cells. We also note that the same inference parameters $\phi_z$ are used as in the non-NTC case. We then derive a corresponding bound

$$\begin{aligned}
\mathcal{L}_{NTC}(x_j^N, x_j^M) = &\mathbb{E}_{q_{\phi_z}(z_j, \mid x_j^N, x_j^M)} \left[\log p_\theta(x_j^N, x_j^M, \mid z_j, t_j = 0)\right] \\
&- D_{KL}(q_{\phi_z}(z_j \mid x_j^N, x_j^M) \| p(z_j))
\end{aligned} \tag{5}$$

By fixing $t_j = 0$ for NTC cells during inference, we prevent the salient variables $t_j$ from capturing any variations for cells with NTC guides. Because the parameters of our background variable inference network $\phi_z$ are shared across both NTC and non-NTC guide cells, our background variables $z$ are thus implicitly encouraged to capture variations shared across control and perturbed cells. In contrast, the salient variables $t$ are free to capture the remaining variations only found in perturbed cells. We assume that all cells are generated independently, so we may then perform inference by maximizing the sums of Equation (4) and Equation (5) across all cells via minibatch gradient ascent similar to standard cLVMs.

When optimizing our final lower bound, the KL divergence terms in Equation (4) and Equation (5) can be computed analytically, as all relevant terms are Gaussian. While our conditional likelihood cannot be computed in closed form, as noted in Section 3 we may instead approximate this term via a pre-trained neural network approximator that allows us to compute estimates of the conditional likelihood in a manner compatible with automatic differentiation; using these approximations we then compute estimates of the expectation terms via the reparameterization trick (Kingma & Welling, 2013). Finally, the perturbation-specific salient means $\mu_c$ are learned as point estimates and optimized along with our model's other parameters.

## 5 RESULTS

To evaluate our proposed method, we applied it to a Perturb-seq dataset originally presented in Norman et al. (2019). In this work the authors studied the effects of CRISPR activation (CRISPRa) perturbations on RNA expression in K562 cells, though only mature mRNA counts were considered in their original work. In contrast, for our analysis we consider both nascent and mature mRNA counts obtained by realignment of this dataset to an intron-annotated reference as done in Chari et al. (2024). As in previous works evaluating the performance of cLVMs (Weinberger et al., 2023; 2024; Lopez et al., 2024), for our analysis we focused on a subset of these perturbations labelled by Norman et al. (2019) as inducing biologically coherent gene programs as a result of perturbation.

### 5.1 ASSESSING REPRESENTATION QUALITY

We began our analysis by assessing whether ContrastiveBiVI's salient latent space successfully isolated perturbation-induced variations from those shared with controls. Based on Norman et al. (2019)'s original analysis, we might hope that cells' latent representations would separate according to their corresponding gene program labels. However, previous analyses of Perturb-seq data have found that confounding sources of variation, due to cell cycle effects, batch effects, or cellular stress responses, may obscure perturbation effects of interest (Papalexi et al., 2021; Weinberger et al., 2023). Indeed, when applying the original biVI model of Carilli et al. (2024) to this dataset, which learns a single latent space that captures all variations in the data, we find that cells exhibit strong separation due to cell cycle in addition to some separation by gene program (**Fig. 2a-b**). On the other hand, we find that ContrastiveBiVI's salient latent representations exhibit clearer separation by known gene programs and exhibit less confounding by cell cycle phase (**Fig. 2c-d**).

To quantify these phenomena, we assessed separation by pathway label via the adjusted Rand index (ARI) and invariance across cell cycle phases via the entropy of mixing. We found that Contrastive-BiVI's salient latent representation achieved substantially stronger performance compared to the biVI model of Carilli et al. (2024) on these metrics. Moreover, to assess the impact of jointly modelling both nascent and mature mRNA counts versus mature counts alone, we also compared against previously proposed generative models designed solely to model mature mRNA counts. In particular, we considered scVI (Lopez et al., 2018) and its corresponding cLVM extension ContrastiveVI (Weinberger et al., 2023).

We found (**Fig. 2e-f**) that ContrastiveBiVI outperformed both of these additional baseline methods in terms of gene program separation, though both ContrastiveBiVI and ContrastiveVI achieved similar mixing across cell cycle phases. Interestingly, we found that the bimodal biVI achieved worse separation of gene programs compared to the mature-mRNA-only scVI model, potentially suggesting the presence of additional confounding factors shared with controls when considering both modalities.

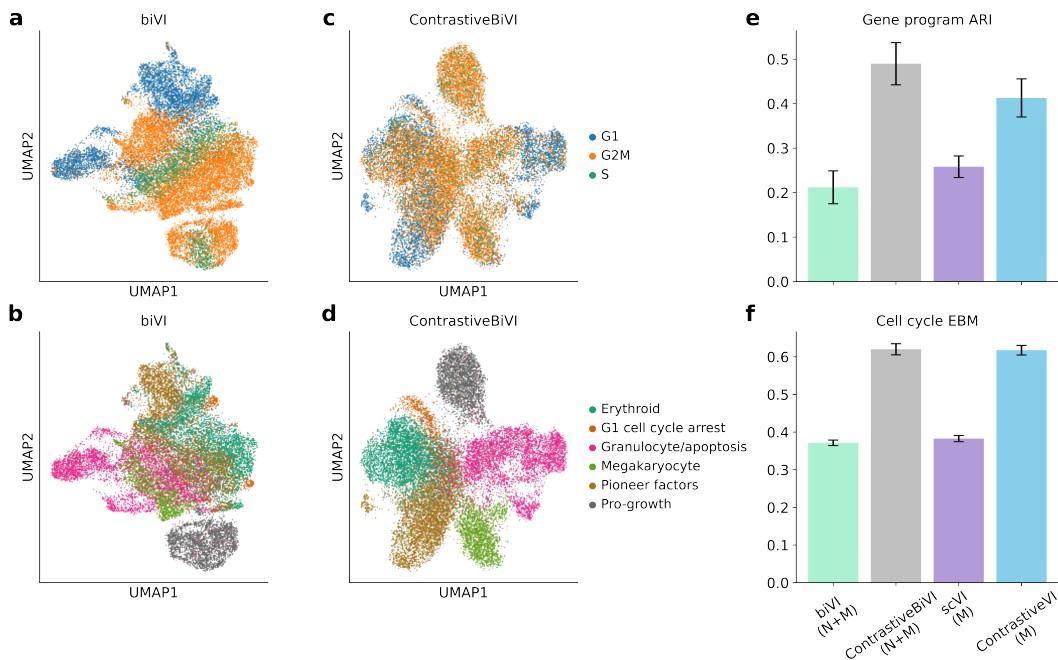

Figure 2: **a-b**, UMAP visualizations of biVI's latent representations for the Norman et al. (2019) dataset coloured by cell cycle phase (**a**) and gene program labels (**b**). **c-d**, UMAP visualizations of ContrastiveBiVI's salient latent space coloured by cell cycle phase (**c**) and gene program labels (**d**). **e-f**, Quantitative assessments of recovery of known perturbation-induced variations (pathway ARI; **e**) and invariance to confounding cell cycle variations (entropy of cell cycle mixing; **f**) for ContrastiveBiVI and baseline method's salient representations. A label of (M) denotes that a model was trained solely on mature mRNA counts, while a label of (N+M) denotes that a method was applied to both mature and nascent counts with a biophysically-inspired generative process.

## 5.2 BIOPHYSICAL DECOMPOSITION OF PERTURBATION-INDUCED DIFFERENTIAL SIGNALS

Having established that ContrastiveBiVI can learn representations that isolate perturbation-induced changes in gene expression, we next assessed whether our biophysically inspired model could provide deeper insights on the differences between the groups of cells highlighted in the model's salient latent space compared to looking at mature mRNA levels alone. To do so, for each gene program label we performed a Bayes factor-based hypothesis test (Boyeau et al., 2023) to identify genes with significant differences in inferred biophysical parameters (namely, burst size $b$ and relative degradation rate $\gamma/k$) between cells corresponding to that program versus controls. As a comparison, we conducted a similar set of tests using scVI to identify genes with differences in observed mature mRNA levels.

Across all gene program labels we found that substantially more genes exhibited significant changes in biophysical parameters as compared to mean spliced mRNA expression (**Fig. 3a**). Indeed, for all programs the set of genes with evidence for a shift in relative degradation rate $\gamma/k$ alone was larger than those exhibiting changes in mean mRNA expression. Notably, we find that only 20.5% of genes detected by scVI overlap with burst size hits, while there is no overlap with relative degradation rate hits. Taken together, these results indicate that our richer biophysical model can capture more subtle regulatory changes than can be deduced from mature mRNA abundance alone. We observed a similar pattern when comparing against ContrastiveVI (**Fig. S2b**).

As one specific example of this phenomenon, when comparing cells labelled with the "Pro-growth" program versus controls, we found that the cell adhesion protein *NECTIN4* did not exhibit a significant change in mean spliced mRNA expression (**Fig. 3b**). On the other hand, this gene demonstrated a pronounced shift in inferred burst size $b$ (**Fig. 3c**) together with a significant shift in inferred degradation rate $\gamma/k$ (**Fig. S1**), suggesting compensatory kinetic changes that leave mean spliced

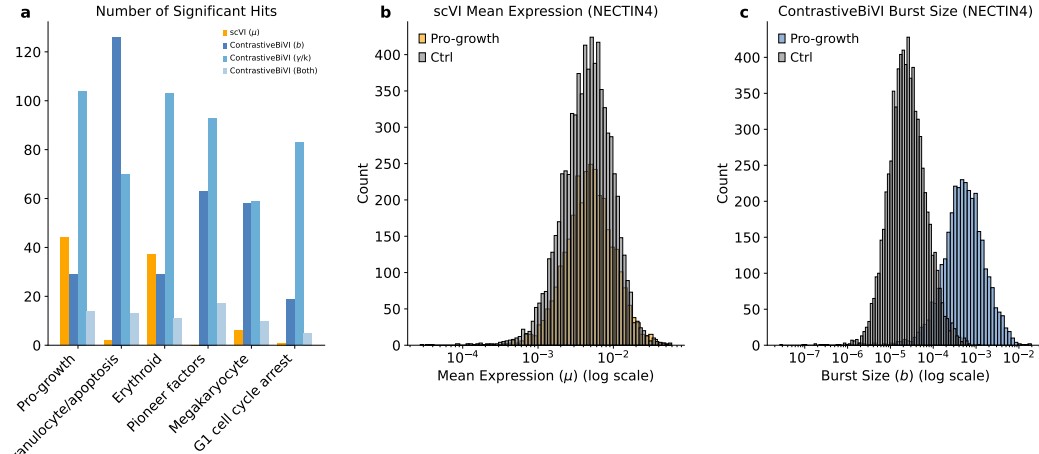

Figure 3: **a**, Number of genes with strong evidence (Appendix E) of program-versus-control shifts in scVI-inferred mean spliced mRNA expression $\mu$ versus ContrastiveBiVI-inferred burst size $b$, relative degradation rate $\gamma/k$, or significant changes in both ContrastiveBiVI parameters. **b**, Inferred mean expression $\mu$ from scVI's likelihood distribution for *NECTIN4* in "Pro-growth" versus control cells (log scale). **c**, Inferred burst size $b$ from ContrastiveBiVI's likelihood parameters for *NECTIN4* in "Pro-growth" versus control cells (log scale).

mRNA largely unchanged. Thus, by resolving these distinct kinetic parameters, ContrastiveBiVI can localize differential evidence to specific modes of transcriptional regulation that may not be discernible through changes in mature mRNA abundance alone.

## 6    CONCLUSION

In this work, we introduced ContrastiveBiVI, a generative model that combines the disentanglement capabilities of contrastive latent variable models with the inherent interpretability of biophysically based models of transcription. We found that ContrastiveBiVI could learn salient representations of bimodal scRNA-seq perturbation datasets that isolate perturbation-induced variations rather than being dominated by common confounders, such as the cell cycle. Moreover, by jointly modelling nascent and mature mRNA with a CME-derived conditional likelihood that explicitly relates the two modalities, ContrastiveBiVI can facilitate a deeper understanding of perturbation effects compared to when considering mature mRNA counts alone. For example, we found that our model could highlight perturbation-induced differences in transcriptional bursting and degradation that are obscured in mature-mRNA-only analyses.

Looking forward, the combination of structured latent variable models combined with biophysically-inspired likelihoods may provide a more principled, mechanistically-aware approach to analyzing perturbation screens. Specifically, this framework enables the classification of perturbations by their kinetic mode of action, distinguishing, for example, between drivers of burst frequency versus modulators of splicing or degradation rates. Future work could extend this resolution by incorporating richer biophysical reaction schemes, such as multi-step splicing or polymerase pausing, into the generative process. Ultimately, by bridging the gap between data-driven representation learning and mechanistic modelling, ContrastiveBiVI presents a grounded approach for dissecting the regulatory mechanisms underlying complex cellular phenotypes.

### MEANINGFULNESS STATEMENT

This work advances meaningful biological representation learning by explicitly grounding latent representations in biophysically interpretable transcriptional processes. ContrastiveBiVI can learn representations that reflect distinct kinetic mechanisms such as transcriptional bursting, splicing, and degradation, rather than treating gene expression as static abundance. By integrating nascent

and mature RNA counts within a contrastive latent variable framework, the learned representations disentangle perturbation-induced effects from shared background variation. This enables representations that are not only predictive, but also mechanistically interpretable, allowing perturbations to be compared and classified by more specific modes of action on transcription.

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

## APPENDIX

## A DERIVATION OF VARIATIONAL BOUNDS FOR CONTRASTIVEBIVI

In this section, we derive the variational bound for cells presented in Section 4. To simplify notation, we denote the concatenation of the two modalities for a cell $i$ as $x_i = [x_i^N, x_i^M]$.

### A.1 DERIVATION FOR PERTURBED CELLS

For cells with non-NTC guides, we assume our variational distribution factorizes as:

$$q_\phi(z_i, t_i \mid x_i) = q_{\phi_z}(z_i \mid x_i) q_{\phi_t}(t_i \mid x_i)$$

where $\phi_z$ and $\phi_t$ denote parameters of inference networks for $z$ and $t$, respectively. Using this variational distribution, and the generative process described in Section 3, the ELBO is derived as follows:

$$
\begin{aligned}
\mathcal{L}(x_i) &= \mathbb{E}_{q_\phi(z_i, t_i \mid x_i)} \left[ \log \frac{p(z_i, t_i, x_i \mid c_i)}{q_\phi(z_i, t_i \mid x_i)} \right] \\
&= \mathbb{E}_{q_{\phi_z}(z_i \mid x_i) q_{\phi_t}(t_i \mid x_i)} \left[ \log \frac{p(x_i \mid z_i, t_i) p(z_i) p(t_i \mid c_i)}{q_{\phi_z}(z_i \mid x_i) q_{\phi_t}(t_i \mid x_i)} \right] \\
&= \mathbb{E}_{q_{\phi_z} q_{\phi_t}} \left[ \log \frac{p(z_i)}{q_{\phi_z}(z_i \mid x_i)} + \log \frac{p(t_i \mid c_i)}{q_{\phi_t}(t_i \mid x_i)} + \log p(x_i \mid z_i, t_i) \right] \\
&= \mathbb{E}_{q_{\phi_z} q_{\phi_t}} \left[ \log p(x_i \mid z_i, t_i) \right] + \mathbb{E}_{q_{\phi_z}} \left[ \log \frac{p(z_i)}{q_{\phi_z}(z_i \mid x_i)} \right] + \mathbb{E}_{q_{\phi_t}} \left[ \log \frac{p(t_i \mid c_i)}{q_{\phi_t}(t_i \mid x_i)} \right] \\
&= \mathbb{E}_{q_{\phi_z} q_{\phi_t}} \left[ \log p(x_i \mid z_i, t_i) \right] - D_{\mathrm{KL}}(q_{\phi_z}(z_i \mid x_i) \| p(z_i)) - D_{\mathrm{KL}}(q_{\phi_t}(t_i \mid x_i) \| p(t_i \mid c_i))
\end{aligned}
$$

### A.2 DERIVATION FOR NON-TARGETING CONTROLS (NTC)

For control cells (e.g., cells that were infected with non-targeting control guides), the generative process assumes no perturbation effect. We impose this inductive bias by constraining the latent variable $t$ to 0. The corresponding ELBO for NTC cells is derived by considering only the reconstruction of the background $z_j$:

$$
\begin{aligned}
\mathcal{L}_{\mathrm{NTC}}(x_j) &= \mathbb{E}_{q_{\phi_z}(z_j \mid x_j)} \left[ \log \frac{p(z_j, x_j \mid t_j = 0)}{q_{\phi_z}(z_j \mid x_j)} \right] \\
&= \mathbb{E}_{q_{\phi_z}} \left[ \log \frac{p(x_j \mid z_j, t_j = 0) p(z_j)}{q_{\phi_z}(z_j \mid x_j)} \right] \\
&= \mathbb{E}_{q_{\phi_z}} \left[ \log p(x_j \mid z_j, t_j = 0) + \log \frac{p(z_j)}{q_{\phi_z}(z_j \mid x_j)} \right] \\
&= \mathbb{E}_{q_{\phi_z}} \left[ \log p(x_j \mid z_j, t_j = 0) \right] - D_{\mathrm{KL}}(q_{\phi_z}(z_j \mid x_j) \| p(z_j))
\end{aligned}
$$

## B  DATASET PREPROCESSING

We used preprocessed spliced and unspliced count matrices generated by Chari et al. (2024) from the original Norman et al. (2019) dataset (GEO accession GSE133344). Following the quality control processing from Norman et al. (2019), we removed doublets and cells with the perturbation label `NegCtrl1_NegCtrl0_NegCtrl1_NegCtrl0`. We further filtered the dataset to retain all cells that received a control treatment and perturbed cells *with a gene program annotation* provided by Norman et al. (2019). Cell cycle phase scores were calculated on the full transcriptome using marker sets defined by Tirosh et al. (2016). Using `scanpy`'s `highly_variable_genes` tool, we subset the dataset to only consider the top 2,000 highly variable genes (HVGs). The final model input consisted of the concatenated spliced and unspliced expression profiles restricted to this gene set for bimodal models and the spliced counts alone for unimodal models.

## C  BASELINES

We benchmarked ContrastiveBiVI against three existing deep generative modelling frameworks in Section 5: biVI (Carilli et al., 2024), scVI (Lopez et al., 2018), and ContrastiveVI (Weinberger et al., 2023). All models were trained on the dataset described in Appendix B. Unimodal methods (scVI, ContrastiveVI) were trained using spliced (mature) counts only, while bimodal methods (biVI, ContrastiveBiVI) were trained using spliced and unspliced counts.

Across all methods, we used the default configuration for the encoder and decoder networks. Specifically, they were implemented as fully connected multilayer perceptrons with a single hidden layer of width 128, ReLU activations, and dropout probability 0.1, with batch normalization enabled and layer normalization disabled.

For methods that learn a single latent space, we treat that latent representation as the quantity of interest when computing representation quality metrics. For contrastive methods, we evaluate the learned salient representation. All quantitative evaluations are reported as averages over five random seeds.

Unless otherwise stated, all models were trained for 100 epochs with batch size 128 using Adam with a learning rate of $10^{-3}$ and weight decay of $10^{-6}$, and a linear KL warmup schedule. All analyses were performed using `scanpy` v1.11.5 (Wolf et al., 2018) and `scvi-tools` v1.4.1 (Gayoso et al., 2022).

### C.1  BIVI

biVI (Carilli et al., 2024) is a biophysically grounded latent variable model that jointly models nascent and mature transcript counts using a mechanistic transcriptional likelihood. Let $x_i^N, x_i^M \in \mathbb{N}^G$ denote nascent and mature counts for cell $i$. The model introduces a latent variable $z_i \in \mathbb{R}^d$ and decodes it into modality-specific normalized expression frequencies,

$$z_i \sim \mathcal{N}(0, I_d), \qquad (\rho_i^N, \rho_i^M) = f_\eta(z_i), \qquad \sum_{g=1}^{G} \rho_{ig}^N = \sum_{g=1}^{G} \rho_{ig}^M = 1.$$

Given a cell-specific library size factor $\ell_i$, the observed count pairs $(x_{ig}^N, x_{ig}^M)$ are modelled using the steady-state distribution of a bursty transcription chemical master equation (CME), parameterized by gene-specific dispersion parameters and modality-specific mean scales $(\ell_i \rho_{ig}^N, \ell_i \rho_{ig}^M)$.

Following Carilli et al. (2024), the CME likelihood is evaluated via a frozen neural approximation that maps biophysical parameters to a finite mixture of negative binomial components, enabling amortized variational inference under a standard ELBO objective.

In our experiments, we set the latent dimensionality to $d = 10$ and used the neural CME approximator of Carilli et al. (2024). Library size was treated as observed.

### C.2  SCVI

scVI (Lopez et al., 2018) is a deep generative model for single-cell RNA sequencing that learns a low-dimensional representation of transcriptomic variation from observed counts. For mature counts

$x_i^M \in \mathbb{N}^G$, scVI introduces a latent variable $z_i \in \mathbb{R}^d$ and decodes it into normalized expression frequencies $\rho_i \in \Delta^{G-1}$,

$$z_i \sim \mathcal{N}(0, I_d), \qquad \rho_i = f_\eta(z_i), \qquad \sum_{g=1}^{G} \rho_{ig} = 1.$$

Observed counts are modelled using an overdispersed count likelihood. In our experiments, we used the standard zero-inflated negative binomial likelihood with gene-level dispersion and observed library size.

We use scVI as a baseline for representation learning from mature mRNA counts only. The latent dimensionality was set to $d = 10$, and we used the standard `scvi-tools` implementation with default settings unless otherwise stated.

### C.3 CONTRASTIVEVI

ContrastiveVI (Weinberger et al., 2023) extends scVI to the contrastive setting by explicitly disentangling variation shared between control and perturbed cells from perturbation-specific variation. Let $x_i^M \in \mathbb{N}^G$ denote mature counts and let $c_i$ denote a perturbation label with a designated control population. The model introduces background latents $z_i \in \mathbb{R}^{d_z}$ and salient latents $t_i \in \mathbb{R}^{d_t}$, which are decoded jointly into normalized expression frequencies,

$$z_i \sim \mathcal{N}(0, I_{d_z}), \qquad t_i \sim \mathcal{N}(0, I_{d_t}), \qquad \rho_i = f_\eta(z_i, t_i), \qquad \sum_{g=1}^{G} \rho_{ig} = 1.$$

The contrastive inductive bias is imposed by clamping salient variables to zero for control cells, such that control cells are reconstructed without salient variation. Observed counts are modelled using a zero-inflated negative binomial likelihood analogous to scVI.

In our experiments, ContrastiveVI serves as a contrastive baseline operating on mature mRNA counts only, allowing us to isolate the effect of contrastive inference from that of explicit biophysical modelling. We set $d_z = 10$ and $d_t = 10$ and use the standard `scvi-tools` implementation. The control population consisted of non-targeting control cells, and the target population consisted of all non-control perturbations.

## D  METRICS

We evaluate learned representations by assessing two complementary properties: (i) their ability to recover known perturbation-induced structure, and (ii) their invariance to dominant confounding sources of variation shared across conditions.

All metrics reported in Fig. 2e–f are computed using perturbed cells only. Let $r_i \in \mathbb{R}^d$ denote the representation used for the evaluation for cell $i$. For all models, we use posterior mean latent representations. For contrastive models, we use the posterior mean of the salient latent variables.

### D.1  ADJUSTED RAND INDEX (ARI)

To quantify how well a representation captures known perturbation structure, we measure agreement between clustering induced in representation space and reference gene program annotations using the adjusted Rand index (ARI).

Let $\{\ell_i\}_{i=1}^n$ denote the gene program labels for the $n$ perturbed cells included in the evaluation. We cluster the corresponding representations $\{r_i\}_{i=1}^n$ using $k$-means, with the number of clusters $k$ set to the number of unique non-control gene programs (six in our experiments), yielding inferred cluster assignments $\{\hat{\ell}_i\}_{i=1}^n$.

Denoting by $n_{ij}$ the number of cells assigned to reference label $i$ and inferred cluster $j$, and defining $a_i = \sum_j n_{ij}$ and $b_j = \sum_i n_{ij}$, the ARI is given by

$$\mathrm{ARI} = \frac{\sum_{i,j} \binom{n_{ij}}{2} - \left[ \sum_i \binom{a_i}{2} \sum_j \binom{b_j}{2} \right] / \binom{n}{2}}{\frac{1}{2} \left[ \sum_i \binom{a_i}{2} + \sum_j \binom{b_j}{2} \right] - \left[ \sum_i \binom{a_i}{2} \sum_j \binom{b_j}{2} \right] / \binom{n}{2}}.$$

ARI corrects for chance agreement between clusterings, with values closer to one indicating stronger alignment between known gene program structure and the separation induced by the learned representation.

## D.2 Entropy of Mixing

To assess invariance of learned representations with respect to cell cycle phase, we compute an entropy-of-mixing score over local neighbourhoods in the representation space. This metric quantifies the extent to which cells from different cell cycle phases are intermixed locally, as would be expected if the representation is insensitive to this confounding source of variation.

Let $g_i \in \{1, \ldots, c\}$ denote the cell cycle phase label for cell $i$, where $c$ is the number of unique phase labels (three in our experiments). For each cell $i$, let $\mathcal{N}_k(i)$ denote its $k$ nearest neighbours in representation space under Euclidean distance, excluding the query cell itself. We use $k = 50$.

We define the empirical neighbourhood proportions

$$p_{i\ell} = \frac{1}{|\mathcal{N}_k(i)|} \sum_{j \in \mathcal{N}_k(i)} \mathbb{1}\{g_j = \ell\}, \qquad \ell \in \{1, \ldots, c\}.$$

The entropy of mixing for cell $i$ is then

$$H_i = -\sum_{\ell=1}^{c} p_{i\ell} \log p_{i\ell},$$

and we report the average of $H_i$ over all perturbed cells, where higher entropy values indicate stronger local mixing of cell cycle phases and therefore greater invariance of the learned representation to cell cycle-associated variation.

## E  Bayes factor-based Significance Testing

Differential analyses were performed using Bayes factor-based hypothesis testing for scVI, ContrastiveVI, and ContrastiveBiVI, but with model-specific implementations. For scVI and ContrastiveVI, we use the `change` mode of the differential testing procedure implemented in `scvi-tools`, which performs Bayesian hypothesis testing on an effect-size random variable defined from posterior samples (Boyeau et al., 2023). In this setting, differential signal is defined by an effect-size threshold $\delta$, with the alternative hypothesis corresponding to log-fold changes lying outside the interval $[-\delta, \delta]$. Bayes factors are computed from Monte Carlo estimates of the posterior probability of an effect exceeding $\delta$ in either direction, using the dominant tail probability.

For ContrastiveBiVI, we adapt the Bayes factor-based differential testing procedure introduced by Carilli et al. (2024), which similarly defines differential signal through a log-fold change threshold and estimates Bayes factors by aggregating posterior samples across groups. This procedure is applied to the mechanistic parameters inferred by our model.

In practice, interpreting Bayes factors requires selecting model- and readout-specific decision thresholds, since their numerical scale is not invariant across posterior parameterizations. For scVI and ContrastiveVI, we use a log-fold change threshold of $\delta_\mu = 0.2$ and a Bayes factor threshold of $\mathrm{BF}_\mu > 2$, corresponding to the settings described in Gayoso et al. (2021). For ContrastiveBiVI, we use an effect-size threshold of $\delta_{\mathrm{kin}} = 1$ and a Bayes factor threshold of $\mathrm{BF}_{\mathrm{kin}} > 3$, following the biVI differential testing framework (Carilli et al., 2024) while adopting a more conservative evidence cutoff (Boyeau et al., 2023).

## F  Supplementary Results for Differential Expression Analysis

Fig. S1 shows the complementary shift in relative degradation rate for *NECTIN4*, while Fig. S2 extends the comparison to ContrastiveVI, a contrastive latent variable model that also operates on mean spliced mRNA expression.

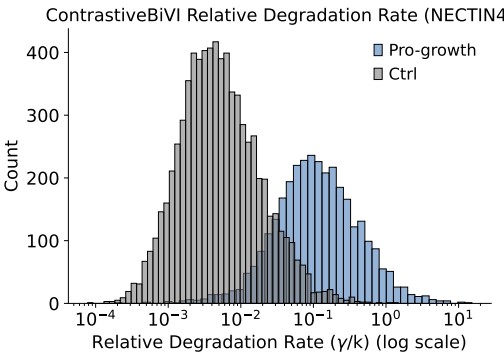

Figure S1: Inferred relative degradation rate $\gamma/k$ from ContrastiveBiVI's likelihood parameters for *NECTIN4* in "Pro-growth" versus control cells (log scale).

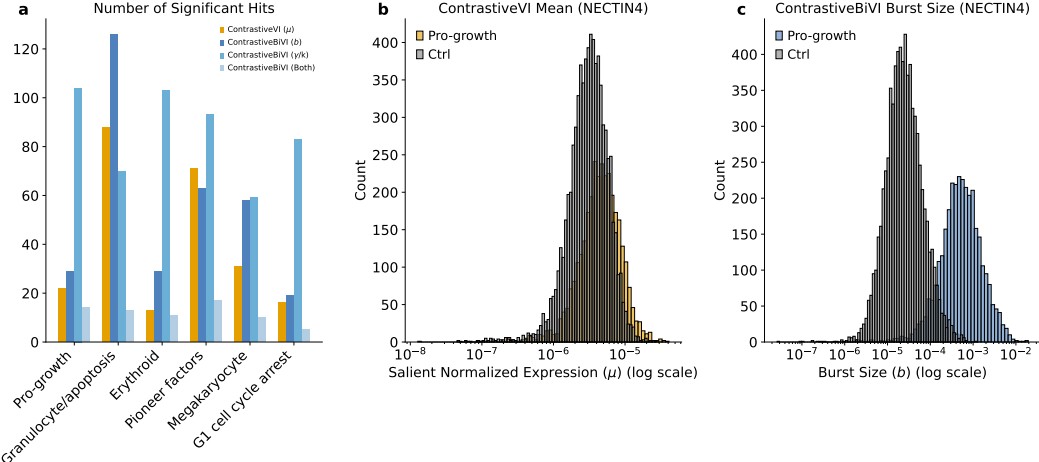

Figure S2: **a**, Number of genes with strong evidence (Appendix E) of program-versus-control shifts in ContrastiveVI-inferred mean spliced mRNA expression $\mu$ versus ContrastiveBiVI-inferred burst size $b$, relative degradation rate $\gamma/k$, or significant changes in both ContrastiveBiVI parameters. **b**, Inferred mean expression $\mu$ from ContrastiveVI's likelihood distribution for *NECTIN4* in "Pro-growth" versus control cells (log scale). **c**, Inferred burst size $b$ from ContrastiveBiVI's likelihood parameters for *NECTIN4* in "Pro-growth" versus control cells (log scale).

