# OpenReview forum: "Exploring Perturbation Effects on Transcriptional Dynamics with ContrastiveBiVI"
_ICLR.cc/2026/Workshop/LMRL — ICLR 2026 Workshop LMRL Poster_

### Official Review · Reviewer_hmA7 · 2026-02-12
**Good mechanistic integration, missing thorough evaluations.**

**Rating:** 7
**Confidence:** 5

**Review:**

**1. Summary**

This paper proposes ContrastiveBiVI, a generative model for single-cell CRISPR perturbation data that combines contrastive latent variable models with a biophysically grounded likelihood derived from chemical master equations. The model jointly analyzes nascent and mature mRNA counts using a two-species bursty transcription model and introduces salient latent variables to isolate perturbation-specific variation from shared background effects. The authors evaluate the model on a CRISPR activation dataset and report improved separation of gene programs and increased detection of differential biophysical parameters relative to baseline methods.

**2. Strengths**
* The combination of contrastive latent variable modeling with mechanistically motivated likelihoods is conceptually novel and well aligned with the workshop theme.
* The generative process is clearly described and builds directly on prior work (biVI, ContrastiveVI).
* Joint modeling of nascent and mature RNA is biologically meaningful and potentially powerful.
* The evaluation framework is systematic and includes multiple baselines.
* The paper is generally well written and technically coherent.

**3. Weaknesses**
  * **3.1 Identifiability of Biophysical Parameters**
The mechanistic parameters (burst size (b) and relative degradation rate $(\gamma/k))$ are recovered via deterministic transformations of neural network outputs and learned dispersion parameters (Equation 3). However:
$(\mu^N)$ and $(\mu^M)$ are produced by a flexible neural decoder.
$(\alpha_g)$ is a learned global gene-level parameter.
No identifiability analysis or simulation-based recovery study is provided.
Since kinetic parameters are central to the interpretability claims, it is important to demonstrate that they are recoverable and stable under the model. At present, it is unclear whether the inferred parameters are uniquely supported by the data or primarily shaped by architectural inductive bias.

  * **3.2 Evaluation is Partly Circular**
The model is trained using perturbation labels and evaluated based on separation of gene program annotations derived from the same dataset. This limits the strength of the validation. External validation, held-out perturbations, or predictive evaluation would strengthen the claims.

  * **3.3 Lack of Uncertainty Calibration or Posterior Validation**
The paper relies heavily on Bayes factor-based differential testing, but:
Different effect-size and Bayes factor thresholds are used across models.
No simulation-based calibration is performed.
No posterior predictive checks or goodness-of-fit diagnostics are shown.
Because the conditional likelihood is approximated via a neural solver and inference is amortized, demonstrating calibration and robustness would be important for supporting the mechanistic interpretation claims.

  * **3.4 Increased Differential Hits Do Not Imply Increased Biological Signal**
The authors report substantially more genes with significant shifts in kinetic parameters compared to mean expression. However:
The model has greater flexibility and more parameters, which can increase detection rates.
There is no independent biological validation or replication analysis.
The presented example **(NECTIN4)** is illustrative but anecdotal.
Without external validation or enrichment analysis, it is difficult to conclude that additional discoveries reflect true biological signal rather than model expressivity.

  * **3.5 Limited Experimental Evaluation**
The representation-level evaluation (UMAP, ARI, entropy of mixing) demonstrates improved clustering of known gene programs. However:
UMAP visualizations are qualitative.
ARI evaluates clustering agreement rather than predictive recovery of gene-level perturbation effects.
No held-out or cross-dataset validation is provided.
The differential testing analysis would benefit from independent biological benchmarks.

  * **3.6 No Discussion of Computational Cost**
The method introduces a bimodal likelihood with a neural approximation of the CME and dual latent spaces. The computational overhead relative to scVI or ContrastiveVI is not discussed. Scalability considerations are relevant, especially in an age of ever-growing perturbation dataset sizes.

**4. Relation to Prior Work**

The work is well situated relative to biVI, scVI, and ContrastiveVI. The extension to a contrastive, mechanistically grounded bimodal model is a meaningful conceptual contribution. However, the evaluation does not yet convincingly demonstrate advantages beyond improved clustering and increased detection counts.

**5. Questions for the Authors**

  1. Can the authors provide simulation-based evidence that burst size ($b$) and degradation parameters ($\gamma/k$) are identifiable under this architecture?
  2. How sensitive are the results to the chosen Bayes factor ($BF$) and effect-size thresholds?
  3. Have the authors evaluated predictive performance on held-out perturbations or independent datasets?
  4. What is the computational overhead relative to biVI and ContrastiveVI?

**6. Overall Assessment**

This paper presents a compelling integration of contrastive representation learning with mechanistic transcription modeling. The approach is conceptually novel, technically coherent, and strongly aligned with the goals of the LMRL workshop.
While questions remain regarding identifiability, calibration, and the strength of empirical validation, these pertain to the depth of substantiation rather than the soundness of the framework. The core methodological contribution is clear and timely, and the work is likely to stimulate valuable discussion within the workshop community.

---

### Official Review · Reviewer_itYo · 2026-02-24
**Interesting ideas, contributions are not well motivated, essential benchmarks are missing**

**Rating:** 5
**Confidence:** 5

**Review:**

The authors combine two established VI methods one for biophysical modeling, biVI, and one for disentanglement in genetic screens, ContrastiveVI, to model mRNA splicing dynamics in post perturbation.
Major comments:
The work is incremental and novel contributions relative to ContrastiveVI are not well motivated. In particular, their Bayes Factor-based hypothesis testing is a standard technique for comparison of two groups (e.g. control vs perturbed).  The model does not resolve changes in transcription and splicing kinetics per perturbation for, for example, perturbation trajectory analysis. The NECTIN4 analysis is over all perturbations lacks comparison to ContrastiveVI, and overall no significant performance gain in disentanglement is evident relative to ContrastiveVI.

Minor comments:
- Fig. 2, titles in c-d should be ContrastiveBiVI, please add ContrastiveVI embedding as well. Also may help to swap e and f to match the order in c-d
- Work in incremental w.r.t ConstrastiveVI.  Authors need to present novel applications.
- The authors have taken strong pertubations in Norman. Additional datasets and perturbations with weaker effects should be considered.
- Perturbation scaping (incomplete knockouts) phenomena is not considered in the proposed model merge
- Representation-based results in Fig2 not different from contrastive VI or substantially outperforming contrastiveVI in case of Gene Program ARI
- Fig3: would be helpful to see mean expression for this gene in contrastiveVI
- Ablations or experiments to show BiVI is significantly adding to ContrastiveVI in isolating perturbation-induced changes in gene expression
-  For the NECTIN4 results comparison to ContrastiveVI and causal links to the perturbations is missing, as the analysis is done at the cell subpopulation level.
- It would be helpful if authors explain why is detecting more genes with significant changes in biophysical params is desired?

I'd be happy to revise my score if ContrastiveVI results for NECTIN4 are added showing that ContrastiveVI alone does not detect a change in mean expression of mature mRNA, and authors could provide novel applications/findings enabled by ContrastiveBiVI that is not possible by ContarstiveVI or BiVI alone.

---

### Official Review · Reviewer_TayD · 2026-02-25

**Rating:** 6
**Confidence:** 5

**Review:**

The paper proposes ContrastiveBiVI, a deep generative model designed to analyze single-cell RNA sequencing (scRNA-seq) data from CRISPR genetic perturbation screens. The authors identify a gap in current computational analyses, which primarily focus on mature mRNA abundance and ignore underlying biophysical processes like transcriptional bursting and mRNA splicing. ContrastiveBiVI addresses this by combining contrastive latent variable models (cLVMs) with a biophysically grounded generative model that jointly models nascent and mature mRNA counts.

### Strengths

- Novel Integration: The core idea of merging mechanistic/biophysical modeling with contrastive disentanglement is highly original and addresses a real need in computational biology.

- Biophysical Interpretability: Unlike standard cLVMs that only detect shifts in mean expression, ContrastiveBiVI allows researchers to localize differential perturbation effects to specific kinetic parameters.

- Logical Ablation Baselines: By comparing against scVI (non-contrastive), biVI (non-contrastive), and ContrastiveVI (contrastive), the authors ablate their model to prove that both the bimodal biophysical likelihood and the contrastive architecture are necessary for their results.

### Weaknesses

- Limited Scope of Evaluation: The empirical evaluation relies entirely on a single dataset (the Norman et al. 2019 CRISPRa screen on K562 cells). To convincingly demonstrate the robustness of the method, it should be evaluated across multiple datasets with varying perturbation modalities (e.g., CRISPRi, small molecules). Potentially simple datasets are RPE1 (Replogle 2022) and Hepg2 and Jurkat (Nadig 2024).

- Direct competing approaches are not benchmarked : CPA (Compositional Perturbation Autoencoder) is the gold standard direct competitor, and its absence from the baseline comparisons is a notable weakness in this paper.

---

### Meta-Review · Area_Chair_kZ8D · 2026-02-25

**Recommendation:** Accept (Poster)
**Confidence:** 4

**Metareview:**

Accept

---

### Decision · Program_Chairs · 2026-03-02

**Decision:**

Accept (Poster)

**Comment:**

Please see the meta-review.